# Dysfunctional Mitochondria in the Cardiac Fibers of a Williams–Beuren Syndrome Mouse Model

**DOI:** 10.3390/ijms241210071

**Published:** 2023-06-13

**Authors:** Noura Abdalla, Ester Tobías-Baraja, Alejandro Gonzalez, Gloria Garrabou, Gustavo Egea, Victoria Campuzano

**Affiliations:** 1Department de Biomedical Sciences, School of Medicine and Health Sciences, University of Barcelona, 08036 Barcelona, Spain; nabdalla89@ub.edu (N.A.); gegea@ub.edu (G.E.); 2Department of Internal Medicine, School of Medicine and Health Sciences, University of Barcelona, 08036 Barcelona, Spainggarrabou@ub.edu (G.G.); 3Centro de Investigación Biomédica en Red de Enfermedades Raras (CIBERER), ISCIII, 28029 Madrid, Spain; 4Institut d’Investigacions Biomèdiques August Pi i Sunyer (IDIBAPS), 08036 Barcelona, Spain; 5Center of Medical Genetics, University of Antwerp, 2650 Antwerp, Belgium

**Keywords:** Williams–Beuren syndrome, CD mouse model, mitochondria, ATP synthesis, respiratory chain, cardiac tissue

## Abstract

Williams–Beuren syndrome (WBS) is a rare neurodevelopmental disorder that, together with a rather characteristic neurocognitive profile, presents a strong cardiovascular phenotype. The cardiovascular features of WBS are mainly related to a gene dosage effect due to hemizygosity of the elastin (*ELN*) gene; however, the phenotypic variability between WBS patients indicates the presence of important modulators of the clinical impact of elastin deficiency. Recently, two genes within the WBS region have been linked to mitochondrial dysfunction. Numerous cardiovascular diseases are related to mitochondrial dysfunction; therefore, it could be a modulator of the phenotype present in WBS. Here, we analyze mitochondrial function and dynamics in cardiac tissue from a WBS complete deletion (CD) model. Our research reveals that cardiac fiber mitochondria from CD animals have altered mitochondrial dynamics, accompanied by respiratory chain dysfunction with decreased ATP production, reproducing alterations observed in fibroblasts from WBS patients. Our results highlight two major factors: on the one hand, that mitochondrial dysfunction is probably a relevant mechanism underlying several risk factors associated with WBS disease; on the other, the CD murine model mimics the mitochondrial phenotype of WBS and could be a great model for carrying out preclinical tests on drugs targeting the mitochondria.

## 1. Introduction

Williams–Beuren syndrome (WBS, OMIM 194050) is a rare neurodevelopmental disorder with an estimated prevalence of 1 in 7500–20,000 newborns [1]. It is caused by the heterozygous deletion of 26–28 contiguous genes (1.55–1.83 Mb) on chromosome 7q11.23 [2]; for a review see [3]. Along with a quite characteristic neurocognitive profile and dysmorphic features, WBS individuals present a severe cardiovascular phenotype. High blood pressure and supravalvular ascending aortic stenosis (SVAS) are the most common cardiovascular manifestations of WBS, affecting more than 75% of patients [4]. Severe SVAS often leads to cardiac hypertrophy, increasing the risk of complications such as stroke and sudden death [4,5]. These cardiovascular features of WBS are mainly related to a gene dosage effect due to hemizygosity of the elastin gene (*ELN*), which encodes the elastin protein. Nonetheless, the phenotypic variability among WBS patients indicates the presence of important genetic and/or epigenetic modulators with significant clinical impact on *ELN* deficiency [6]. Mechanistically, reactive oxygen or nitrogen species (ROS and RNS, respectively) levels have been described as strong determinants of hypertension risk and vascular stiffness in WBS patients [7,8]. Both findings have been experimentally supported in murine models of the WBS orthologous region [9,10]. One important source of ROS is the mitochondrion. In fact, mitochondrial dysfunction seems to be associated with numerous cardiac diseases, such as desminopathies [11,12], atherosclerosis, ischemia–reperfusion injury, heart failure, and hypertension, presumably due to insufficient cellular energy production and dysfunctional ROS production [13]. Cardiomyocytes are characterized by their high energy consumption, which is highly dependent on basal mitochondrial activity and their dysfunction appears to be especially sensitive to aberrations of the mitochondrial turnover process [14]. The oxygen consumption rate (OCR), or mitochondrial respiration, can be used to measure mitochondrial bioenergetics [15]. A growing body of evidence suggests that mitochondrial dynamics (MD) contribute to mitochondrial respiration [16,17]. MD is regulated by fusion and fission processes that influence the shape, size, and number of mitochondria [18]. Dynamin-related guanosine triphosphatases (GTPases) regulate mitochondrial fission and fusion events [19]. Mitochondrial fusion is regulated by mitofusin-1 (MFN1) and mitofusin-2 (MFN2), which are anchored in the outer mitochondrial membrane, and by optic atrophy 1 (OPA1), anchored in the inner mitochondrial membrane. Cells lacking MFN1 and MFN2 exhibit fragmented mitochondria, which is associated with reduced mitochondrial respiration [16,20]. Proteolytic processing of OPA1 in the inner mitochondrial membrane is emerging as a critical regulatory step to balance the mitochondrial fusion–fission process. L-OPA1 is required for mitochondrial fusion, while excess accumulation of S-OPA1 accelerates fission. An imbalance between the activities of these groups of proteins leads to deficient or excessive mitochondrial fragmentation, which has been demonstrated in several cardiovascular diseases such as atherosclerosis, reperfusion injury, cardiomyopathy, and cardiac hypertrophy [14]. Additionally, mitochondrial fission is regulated by mitochondrial fission protein 1 (FIS1) and dynamin-related protein 1 (DRP1). Moreover, mitochondria participate in maintaining Ca^2+^ homeostasis; transfer of Ca^2+^ from the endoplasmic reticulum, which serves as a primary cellular Ca^2+^ reservoir, can lead to mitochondrial dysfunction and fragmentation of the organelle [21].

The involvement of mitochondrial dysfunction in WBS pathology has recently been highlighted in WB patients. It was reported that they exhibited several features of mildly accelerated aging, which may be caused by dysfunctional mitochondria, which in turn may subsequently facilitate neurodegenerative hallmarks. The deletion of two genes included in the WBS critical region, *DNAJC30* and *BUD23*, has been associated with mitochondrial dysfunction. DNAJC30 is an auxiliary protein of the ATP–synthase complex. Decreased DNAJC30, whether in tissue from WBS patients or KO mice, resulted in decreased ATP levels, as well as alterations in neuronal morphology, electrophysiology, and mitochondrial function [22]. BUD23 is highly conserved throughout evolution and has recently emerged as a ribosomal RNA methyltransferase, with a role in maintaining mitochondrial oxidative phosphorylation capacity. Indeed, in mice, cardiomyocyte loss of BUD23 greatly impaired mitochondrial ATP generation, leading to dilated cardiomyopathy and premature death. The resulting cardiac phenotype was severely altered, with marked mitochondrial dysfunction [23].

The WBS complete deletion (CD) mouse model exhibits an increased heart weight/body weight ratio due to increased cardiomyocyte cross-sectional area consistent with cardiac hypertrophy [10]. Additionally, transthoracic echocardiography measurements showed that the left ventricular mass and wall thickness in the interventricular septum and posterior wall were significantly increased in CD mice. Consequently, the CD mice showed a slight reduction in their left ventricular ejection fraction [24]. Recently, we demonstrated that an important source of oxidative stress in the cardiovascular system is xanthine oxidoreductase (XOR) [25]. XOR reacts with hypoxanthine forming superoxide (O_2_^−^) as a by-product, which damages mitochondria, leading to bioenergetic dysfunction. O_2_^−^ damages the respiratory chain complexes either directly or via interactions with nitric oxide (NO), generating peroxynitrite that elicits diverse deleterious effects on the mitochondria. Additional mitochondrial damage due to increased ROS/RNS and decreased ATP production via a feedback loop results in left ventricular dysfunction [26].

It is still too early to think of the possible clinical use of mitochondrial fission/fusion regulation. However, some recent encouraging results reported that the administration of the pharmacological mitochondria fission inhibitor mdivi1 in a mouse model mitigated abdominal aortic aneurysms [27]. In an attempt to broaden the range of new therapeutic alternatives in a disease that lacks a defined pharmacological treatment, herein, we study the basal status of mitochondrial function in the CD model to subsequently examine the most suitable therapeutic intervention targeting mitochondria [28,29].

## 2. Results

### 2.1. The Cardiac Fibers of CD Mice Exhibit Deficient Oxygen Consumption

Given the direct involvement of two *WBSCR* genes in mitochondrial function [22,23] and taking into account that CD mice phenocopy the cardiovascular defect of WBS patients [10], we examined whether similar underlying cellular features were present in CD cardiac tissue.

Functional analysis of mitochondrial function in permeabilized cardiac fibers using the OROBOROS microrespirometer system revealed that mitochondrial oxidative phosphorylation (OXPHOS) activity was significantly reduced in the CD mice, while LEAK respiration was unaffected (Figure 1A).

The respiratory control ratio (RCR, calculated here as OXPHOS/LEAK) measures the efficiency of mitochondrial ATP production. The reduction in OXPHOS led to a lower RCR, indicating that the mitochondria from the CD mice were less efficient at producing ATP (Figure 1B). The reduction in ATP production was also validated in primary mouse embryonic fibroblasts (MEFs) by a kinetic luciferase activity assay (Figure 1C). Detailed statistical analyses are shown in Appendix A. 

### 2.2. CD Cardiac Tissue Show Decreased Levels of the OXPHOS Complex

Next, we evaluated if the reduction in ATP production could be due to a loss of mitochondrial mass. The mitochondrial genomic copy number was measured relative to the nuclear genomic copy number. We observed a significant increase in mitochondrial DNA content both in the cardiac tissue and primary MEFs of the CD mice (Figure 2A,B).

Functional differences in the OXPHOS pathway prompted us to search for potential differences within OXPHOS protein complexes. We analyzed the expression of electron transport chain proteins in heart homogenates via Western blotting with an OXPHOS cocktail antibody. Expression patterns of mitochondrial complexes I, II, III, IV, and V, referenced as NADH dehydrogenase (ubiquinone) 1 beta subcomplex 8 (NDUFB8), succinate-ubiquinol oxidoreductase iron sulfur protein (SDHB), ubiquinol cytochrome c reductase core protein II (UQCR2), cytochrome c oxidase subunit 2 (MTCO1), and ATP synthetase F1 complex α subunit (ATP5A1), respectively, were significantly reduced in the CD samples (Figure 2C,D). Detailed statistical analyses are shown in Appendix A.

### 2.3. Altered Mitochondrial Morphology in CD Cardiac Fibers

We next investigated whether the observed mitochondrial dysfunction in the CD cardiomyocytes was accompanied by alterations in mitochondrial morphology and/or dynamics in the heart tissue. Transmission electron microscopy (TEM) analysis of cardiac tissue revealed a marked disorganization of the CD cardiac tissue compared with the highly ordered mitochondrial pattern observed in wild-type (WT) cardiac tissue. In the CD cardiac tissue, the mitochondria were smaller in size with a more circular shape (Figure 3A-top). The CD mitochondria appear to contain structurally similar cristae compared to the wild type (Figure 3A-bottom). In agreement with the TEM observations, confocal analysis of CD MEFs showed a significantly increased mitochondrial density accompanied by a significant reduction in average mitochondrial size (Figure 3B). The value of circularity was significantly closer to 1 in the CD mitochondria, which would signify its dysfunction (Figure 3B). Detailed statistical analyses are shown in Appendix A.

### 2.4. Altered Mitochondrial Dynamics in CD Cardiac Fibers

At first sight, the morphological characteristics of the mitochondria were consistent with an increased mitochondrial fission process, which would predictably cause reduced total mitochondrial function. To test this, we next analyzed the MD-associated protein levels in heart homogenates from CD mice. Western blot analysis showed a significant imbalance between OPA1 isoforms. Likewise, we observed a significant decrease in L-OPA concomitantly with an increase in S-OPA in CD cardiac proteins (Figure 4A). This imbalance was accompanied by increased FIS1 levels (Figure 4B). Finally, we also observed reduced MFN1 levels with no change in MFN2 levels (Figure 4C,D). All the data concur with an increased mitochondrial fission process that accompanies the observed morphological shape of CD mitochondria. Detailed statistical analyses are shown in Appendix A.

## 3. Discussion

Despite all the advances made in the understanding of WBS since its initial description, many unanswered questions remain. These questions focus on three large blocks: molecular mechanisms of the disease, inter-individual variability, and effective treatment strategies. A more complete understanding of the genes and pathways that contribute to human WBS phenotypes would allow clinicians to move beyond mere symptom control, using precisely targeted therapies to improve organ dysfunction and, ultimately, outcomes. Cardiovascular injury is present in most individuals with WBS [4,30,31]. Depending on the location, severity, and timing of onset, the management of the cardiovascular pathology consists of noninvasive or surgical intervention complemented with life-long monitoring. Both environmental factors and genetic modifiers likely contribute to the overall penetrance of specific signs and symptoms in an individual with WBS. It is well known that the risk of hypertension is associated with ROS production via the NOX signaling pathway [9,32]. Mitochondrial dysfunction has been recently indicated as being involved in WBS pathogenesis; however, it has not been extensively explored, which is precisely where our study fits in. Decreased basal respiration and maximal respiratory capacity, increased ROS generation, and decreased ATP synthesis have been demonstrated in primary WBS fibroblasts [22]. Our study reveals two basic new points in the CD model of WBS disease: (i) the confirmation of functional mitochondrial alterations as previously reported in WBS patients [22] and (ii) this model could be the best one for carrying out preclinical tests on drugs targeting the mitochondria.

Herein, we report reduced oxygen consumption (OXPHOS) and lower RCR in permeabilized cardiac fibers from CD mice, indicating that the mitochondria were less efficient at producing ATP. We validated the reduction in ATP production in primary MEFs from CD animals showing a significant reduction in ATP production, correctly mimicking the differences observed in human WBS fibroblasts [22]. Of note, the reduction in the respiration rate correlated with decreased protein expression of the respective mitochondrial complexes. This general reduction could be attributed in part to the haploinsufficiency of BUD23, a highly conserved ribosomal RNA methyltransferase that promotes the translation of mitochondrial proteins. The mitochondrial respiratory capacity of CD cardiac fibers was very similar to that of cardiac tissues of BUD23-depleted cardiomyocytes. However, no significant difference in mitochondrial respiratory capacity across any mitochondrial state was observed in heart homogenates from *Bud23+/−* mice [23]. This more severe effect observed in our model could be attributed to the accompanying *Dnajc30* haploinsufficiency, an auxiliary protein of the ATP synthase complex. Cultured primary neurons of *Dnajc30* KO mice showed a significantly reduced basal oxygen consumption rate, and ATP production was decreased in mitochondria extracted from *Dnajc30* KO mouse cortices without any apparent changes in the OXPHOS super complex [22].

Decreased ATP synthesis efficiency has been associated with a reduction in mitochondrial size [33,34]. Smaller, more circular mitochondria were clearly observed in the cardiac tissue of the CD animals via TEM. A similar mitochondrial circular shape can be seen in both DNAJC30 and BUD23 simple deletion models [22,23]. Unlike in the latter, the authors point out the absence of electron-dense spheroid inclusions, which are not appreciable in our model [23]. The analysis of mitochondrial shape in CD-derived primary MEFs by confocal microscopy agrees with the TEM observations of CD cardiac tissue, with the presence of smaller, more circular mitochondria. Likewise, we observed greater mitochondrial density, which is also in accordance with the increased mitochondrial genomic copy number observed.

The mitochondrion is a highly dynamic organelle that constantly fuses and divides, thus maintaining its homeostasis. The disruption of the normal mitochondrial fusion process has been shown to lead to smaller fragmented mitochondria, causing impaired cellular respiration [16,34]. In addition, it has been associated with a variety of neurodegenerative and cardiovascular diseases [35,36]. The analysis of MD-associated proteins in the cardiac tissue of CD mice revealed a shift toward fission, similar to that observed in a variety of cardiovascular diseases [37]. This was caused by the reduction in L-OPA1 forms, which mediate the fusion process, with the concomitant accumulation of S-OPA1 forms, indicative of mitochondrial fragmentation, together with the reduction in MFN1 and increased FIS1 protein levels.

We conclude that in our CD WBS mouse model, cardiac fibers contain abnormal mitochondria, both in structure and function, leading to a reduced respiratory capacity and ATP synthesis, reminiscent of fibroblasts derived from WBS patients. Mitochondrial dysfunction is likely a relevant mechanism underlying several risk factors associated with WBS disease. This observation allows us to assay new pharmacological therapies targeting mitochondrial dysfunction and their impact on cardiac hypertrophy and hypertension. Pharmacological intervention, often resulting in transient and partial inhibition of the activated fission pathway, appears to be the most promising approach [38,39,40]. In the case of WBS, HO-1 could be a good candidate since it can prevent mitochondrial fragmentation by increasing *Mfn1/2* expression and decreasing *Fis1* expression [41]. Finally, it would also be of great interest to examine whether mitochondrial activity is also dysregulated in vascular smooth muscle cells of the characteristic stenotic aorta.

## 4. Materials and Methods

### 4.1. Animal Maintenance

CD mice, a WBS murine model carrying a 1.3 Mb heterozygous deletion spanning from *Gtf2i* to *Fkbp6*, were obtained as previously described [10]. All of the mice were maintained in a 97% C57BL/6J background. Genomic DNA was extracted from a mouse ear punch to perform genotyping via PCR with appropriate primers (Appendix A), as previously described [42]. The animals were housed under standard conditions in a 12 h dark/light cycle with access to food and water/treatment ad libitum. All procedures involving animals were performed in compliance with the National Institutes of Health’s Guide for the Care and Use of Laboratory Animals and approved by the local animal care committee of the Universitat de Barcelona (EB-310/22) in accordance with European (2010/63/EU) and Spanish (RD53/2013) regulations for the care and use of laboratory animals.

### 4.2. Cell Lines and Culture Conditions

Mouse embryonic fibroblasts (MEFs) were extracted following a previously described protocol [10]. The cells were cultured in humidified 5% CO_2_ at 37 °C in Dulbecco’s modified Eagle medium (DMEM) supplemented with Fetal bovine serum (FBS, 10%) and 1% antibiotics (penicillin and streptomycin). The cells were washed with phosphate-buffered saline (PBS) to remove media, and trypsin was used for passaging. Culture media were replaced every two days, and the cells were passaged upon reaching 70–80% confluency.

### 4.3. Assessment of Mitochondrial Function Using OROBOROS

Of the three distinct experimental preparations available (isolated mitochondria, permeabilized fibers, and tissue homogenates), we used permeabilized cardiac fibers for mitochondrial assessment. Ventricular tissue (~2–3 mg) was weighed and transferred to 1.5 mL of ice-cold MiRO5 medium (in mM: EGTA 0.5, MgCl_2_ 1.4, taurine 20, KH_2_P0_4_ 10, HEPES 20, BSA 1%, K-MES 60 mM, sucrose 110 mM, pH 7.1, adjusted with 5 N KOH). The cardiac fibers were permeabilized with 50 µg/mL saponin before loading on an Oroboros Oxygraph 2 k high-resolution respirometry system (Oroboros Instruments, Innsbruck, Austria) for measurement of mitochondrial respiration. Two identical respiration chambers (A and B), at the same temperature, were run in parallel for each experiment.

Three parameters are commonly used to assess mitochondrial function [15,43]. Firstly, OXPHOS capacity is the respiratory capacity of mitochondria in the ADP-activated state of oxidative phosphorylation (saturating concentrations of ADP, inorganic phosphate, oxygen, and defined substrates). Secondly, there is the LEAK respiration rate, which represents mitochondrial respiration that occurs in the absence of ATP generation, mainly to compensate for proton leak across the mitochondrial inner membrane. Thirdly, there is RCR (calculated here as OXPHOS/LEAK), which measures the degree of coupling between oxidation and phosphorylation or, in other words, the efficiency of mitochondrial ATP production. OXPHOS and LEAK were measured in the presence of the omplex I substrates pyruvate and malate (electron transfer through complexes I–IV) or complex I+II substrates (addition of succinate). Additionally, respiratory flux with electron transfer through complex IV alone was measured via the addition of the electron donor tetramethyl-phenylene-diamine (TMPD). The protocol used to measure these parameters was adapted from Pesta and Gnaiger (2012) [43]. Briefly, pyruvate (10 mM), malate (2 mM) and glutamate (20 mM) were added as carbon substrates to spark the citric acid cycle. Under these conditions, mitochondria are in LEAK respiration with CI substrates in the absence of adenylates. OXPHOS with CI substrates was achieved through the addition of saturating levels of ADP (2 mM). Following steady-state conditions, succinate (10 mM) was added to achieve OXPHOS with CI+CII substrates. Rotenone (0.5 µM) was then added to achieve ETS with CII substrates and antimycin A (5 mM) to block complex III and measure background non-mitochondrial residual oxygen consumption (ROX). OXPHOS through complex IV alone was assessed by adding the electron donor TMPD (0.5 mM). To avoid its oxidation, ascorbate (2 mM) was added prior to TMPD injection.

### 4.4. ATP Measurements

To measure the ATP content in MEFs, 1.0 × 10^4^ cells were plated in triplicate wells of white 96-well plates. After 12 h to re-establish homeostasis, the cells received fresh media, were incubated for 4 h, and then analyzed by CellTiter-Glo 2.0 following the manufacturer’s protocol. Duplicates were also set up in clear 96-well plates to confirm cell adherence with subsequent DAPI staining for standardization. A standard ATP curve (10^−12^–10^−3^ M) was used for all ATP experiments.

### 4.5. Western Blotting

Frozen hearts were homogenized in RIPA buffer containing protease inhibitors (2 mM phenylmethylsulphonyl fluoride, 10 g/L aprotinin, 1 g/L leupeptin, and 1 g/L pepstatin) and phosphatase inhibitors (2 mM Na_3_VO_4_ and 100 mM NaF). Protein concentration was determined using a Dc protein assay kit (Bio-Rad, Bekerley, CA, USA). Nitrocellulose blotting membranes (Amersham Protran, Amersham, UK) were blotted overnight at 4 °C with the following primary antibodies: total OXPHOS human antibody cocktail (Abcam, Cambridge, UK; 1:5000 dilution), TOM20 (Santa Cruz Biotechnology, Santa Cruz, CA, USA; 1:5000), OPA1 (BD Transduction Laboratories, Madrid, Spain; 1:1000), MFN1, MFN2 and FIS1(Cell Signaling, Danvers, MA, USA; 1:1000). Thereafter, the membranes were washed and incubated with the appropriate HRP-conjugated secondary antibody (Promega, Madrid, Spain; 1:3000), and the reaction was finally visualized with Western Blotting Luminol Reagent (Santa Cruz Biotechnology, Santa Cruz, CA, USA). The band intensity of each protein was quantified by Image j-win64.

### 4.6. qPCR Experiments

qPCR was performed on MEFs using a SYBR Green Ready Master Mix in the ViiA7 Real-Time PCR System (Applied Biosystems, Foster City, CA, USA) according to the manufacturer’s instructions. Raw data were obtained using ViiA7 Software v1.2 (Applied Biosystems). The comparative C_T_ (ΔΔC_T_) method was used for analysis. Semi-qPCR was used for cardiac tissue. The amplification of genomic fragments was used as a control for relative DNA quantification. A reagent-only negative control sample was included in each run. Five MEF experiments and six mice per genotype per cardiac tissue, with three replicates per sample, were performed and analyzed. See Appendix A for the primer sequences.

### 4.7. Analysis of Mitochondrial Morphology

Immunocytochemistry and confocal microscopy analysis were performed as previously described [44] with minor modifications. Briefly, the MEFs were fixed in 4% paraformaldehyde (Electron Microscopy Science EMS, Hatfield, PA, USA) in PBS. Subsequently, the cells were permeabilized with 0.1 M Glycine and 0.1% saponin and then with 1% bovine serum albumin in phosphate-buffered saline for 1 h. The specimens were incubated with primary antibody TOM20 (anti-mouse SC-17764, 1:250, Santa Cruz Biotechnology) overnight at 4 °C. Thereafter, the samples were incubated with goat-AlexaFluo 488 (anti-mouse, 1:600; Thermo Fisher Scientific, Waltham, MA, USA; A11001, Ex/Em: 443/518). The nuclei were stained with the Hoechst 33258 (1:10,000; Molecular Probes, Life Technologies Ex/Em: 352/455), and the cell were visualized with CellMask (1:15,000, Thermo Fisher Scientific, Waltham, MA, USA; H32713 Ex/Em: 545/570). Immunofluorescence was analyzed by confocal microscopy using a Leica TCS SP5 laser scanning spectral confocal microscope (Leica Microsystems, Heidelberg, Germany). Confocal images were obtained using a 63× numerical aperture objective with a 3× digital zoom and standard pinhole. For each cell, the entire three-dimensional stack of images from the ventral surface to the top of the cell was obtained by using the Z drive in the Leica TCS SP5 microscope. Quantitative analyses of mitochondrial morphology were performed with Image j-win64 software (NIH, Bethesda, MD, USA) and a specifically designed macro [45].

### 4.8. Transmission Electron Microscopy (TEM)

After dissection, the hearts were cut into small pieces and fixed with 2.5% glutaraldehyde and 2% paraformaldehyde in 0.1 M phosphate buffer and postfixed with 1% OsO_4_, dehydrated with ethanol, and embedded in Spurr resin before being sectioned with a Leica ultramicrotome UC7 (Leica Microsystems). Ultrathin sections (50–70 nm) were stained with 2% uranyl acetate for 10 min, a lead-staining solution for 5 min, and observed using TEM with a JEOL JEM-1010 fitted with a Gatan Orius SC1000 (model 832) digital camera to seek abnormal organelle structures [46].

### 4.9. Statistical Analysis

Prior to statistical analyses, all data were analyzed by the Shapiro–Wilk test to confirm normality. All data are presented as mean ± the SEM. Values were considered significant when *p* < 0.05. GraphPad Prism 9 software was used for obtaining all statistical tests and graphs.

## Figures and Tables

**Figure 1 ijms-24-10071-f001:**
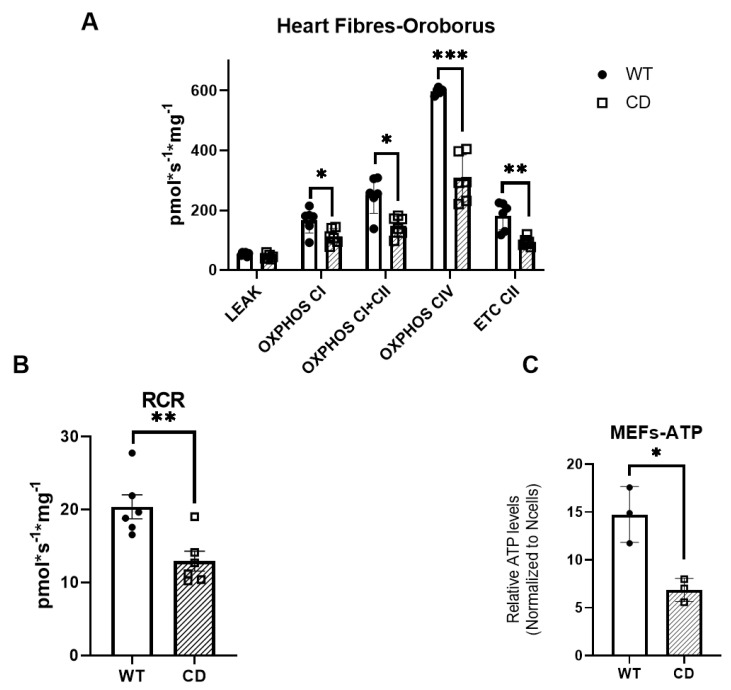
CD mice cardiac fibers exhibit reduced oxygen consumption and ATP production (**A**). Mitochondrial respiration was measured in permeabilized cardiac fibers (n = 5–6). Oxygen consumption was measured under various mitochondrial states and normalized to fiber weight: LEAK conditions (malate, pyruvate, and glutamate as substrates); OXPHOS (routine respiration rate) through complex I, I + II, II, or IV); (**B**) respiratory control ratio (RCR, measure of mitochondrial efficiency). (**C**) ATP production in primary CD and WT MEFs. The cells were seeded and analyzed with the Luciferase assay. ATP levels were normalized to their respective cell numbers. n = 3 experiments with three replicates each. The data were analyzed using a multiple (**A**) or unpaired *t*-test (**B**,**C**) and are represented by the mean ± SD; statistical significance was set at a threshold of * *p* < 0.05; ** *p* < 0.01; *** *p* < 0.001.

**Figure 2 ijms-24-10071-f002:**
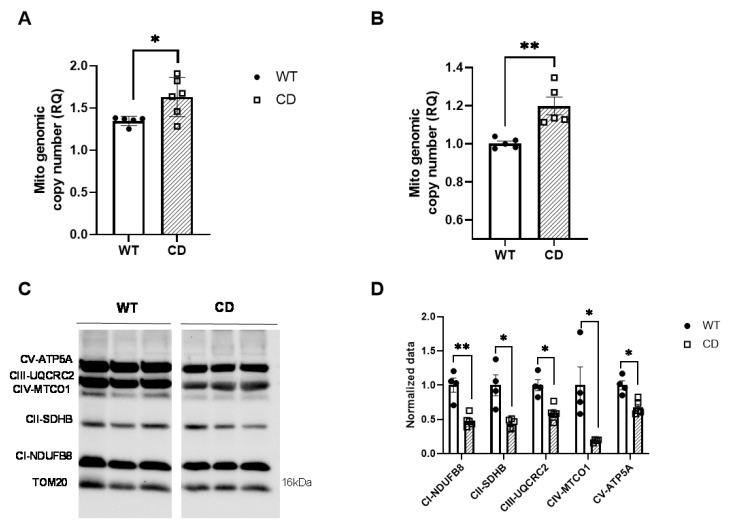
CD mice show reduced levels of OXPHOS protein complexes. The mitochondrial copy number was measured relative to the genomic copy number in cardiac tissue (**A**) and MEFs (**B**). (**C**) Representative Western blot of mitochondrial OXPHOS respiratory complex protein levels. TOM20 was used as a loading control. An antibody cocktail comprising the following subunits of respiratory complex proteins was used: NADH dehydrogenase (ubiquinone) 1 beta subcomplex 8 (NDUFB8; complex I), succinate dehydrogenase complex, subunit B, iron sulfur (SDHB/Ip; complex II), ubiquinol-cytochrome c reductase core protein II (UQCR2; complex III), ATP synthase 5A (ATP 5A, Complex V), and cytochrome c oxidase subunit 2 (COXII; complex IV). (**D**) Quantitative analysis of the protein levels of each of the aforementioned subunits, respectively. The intensity of each protein was normalized to TOM20 in the same sample. The data were analyzed using an unpaired (**A**,**B**) or multiple comparisons (**D**) *t*-test following the Holm–Šídák method and are represented by mean ± SD; n= 4–6 mice per group. The cells were analyzed in n = 3 experiments with three replicates each. Statistical significance was set at a threshold of * *p* < 0.05; ** *p* < 0.01.

**Figure 3 ijms-24-10071-f003:**
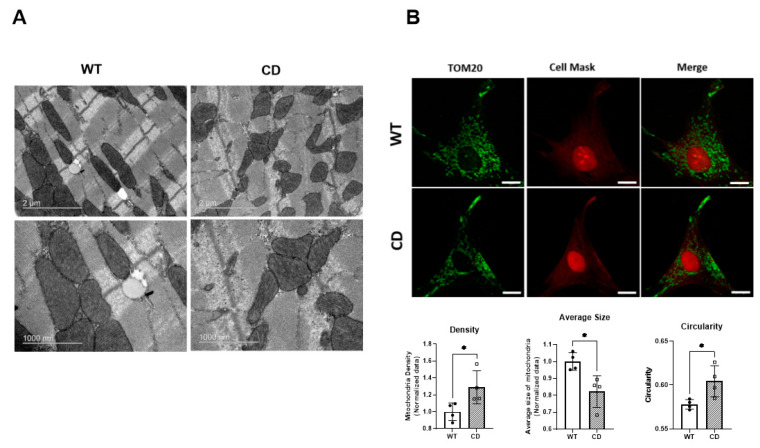
The mitochondrial structure and dynamics were altered in the CD cardiac tissue. (**A**) Representative TEM images of mitochondria distribution along cardiac fibers. Scale bars = 2 μm (**top**) and 0.1 μm (**bottom**), respectively. (**B**) (**Top**), representative confocal images of mitochondria distribution in MEFs (scale bars = 10 μm) showing increased mitochondrial density with a reduced average size and increased circularity (**bottom**). The value of circularity ranges between 0 and 1, with the less functional mitochondria closer to 1. The cells were analyzed in n = 4 experiments with three replicates each. The data were analyzed using an unpaired *t*-test and are represented by the mean ± SD. Statistical significance was set at a threshold of * *p* < 0.05.

**Figure 4 ijms-24-10071-f004:**
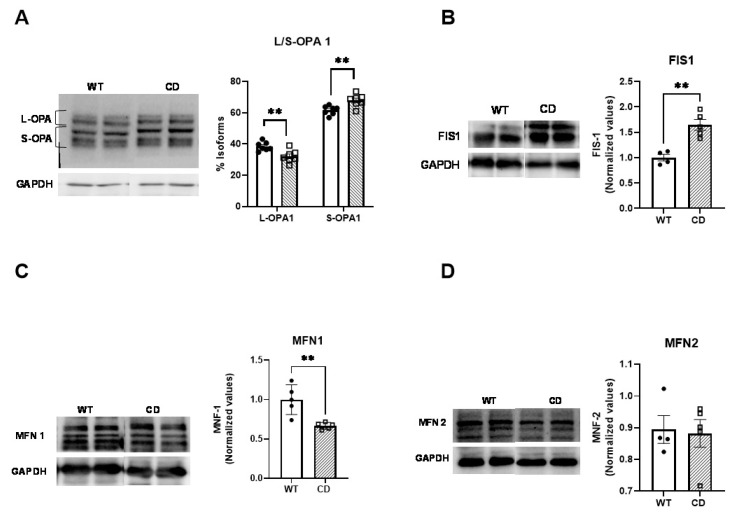
Reduced expression levels of proteins controlling mitochondrial dynamics. (**A**) Quantification (**right**) and representative Western blot (**left**) of OPA 1; (**B**) fission protein FIS1 and fusion proteins, (**C**) MFN1, and (**D**) MFN2. GAPDH was used as a loading control. The data were analyzed using multiple comparisons (**A**) or an unpaired (**B**–**D**) *t*-test following the Holm–Šídák method and represented by the mean ± SD; n = 4–5 mice per group. The cells were analyzed in n = 4–5 mice/genotype. Statistical significance was set at a threshold of ** *p* < 0.01.

## Data Availability

Not applicable.

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
