# Peer review of "Dysfunctional Mitochondria in the Cardiac Fibers of a Williams–Beuren Syndrome Mouse Model"

_ijms, 2023, doi:10.3390/ijms241210071_

Round 1

Reviewer 1 Report

The authors have investigated mitochondrial dysfunction in Williams-Beuren-Syndrome. The manuscript is well written and needs only some minor corrections.

1.) Please indicate the prevalence of WBS in the introduction.

2.) Summarize the genetic locus and the deleted genes explained in line 34/35 in a schematic figure and include this in the introduction.

3.) Please add genetic cardiomyopathies and especially desminopathies to the list of cardiac diseases associated with mitochondrial dysfunction. For example, it is known, that desminopathy, cause by desmin variants, is also associated with mitochondrial dysfunction. Please add therefore this information including a relevant reference.

4.) Rephrase sentence in Line 98/99. This sounds strange.

5.) Explain all abbreviations like FBS or DMEM in the revised manuscript.

6.) Please present all data as mean with standard deviation SD instead of SEM. Larger error bars are not a problem.

7.) Please specify the statistic test, which you used. Non-parametric or parametric tests?

8.) Figure 3B: Scale bars are missing and not explained.

9.) Please specify the exact membranes used for Western blotting.

10.) Please indicate the species used for secondary antibodies?

11.) Please include the oligonucleotides used for qRT-PCR.

12.) Please include the exact excitation and emission wavelengths used for confocal microscopy.

However, I am optimistic that the authors can fixed the points in a minor revision. Good luck!

Minor corrections by a native speaking editor might be necessary.

Author Response

The authors have investigated mitochondrial dysfunction in Williams-Beuren-Syndrome. The manuscript is well written and needs only some minor corrections.

Thanks to the reviewer for their contributions that help improve the quality of this work

  1. Please indicate the prevalence of WBS in the introduction.

The prevalence of Williams Syndrome is indicated in the second sentence of the Introduction (line 36). Corresponds to reference 1, which we have repositioned into the text for clarity.

2.) Summarize the genetic locus and the deleted genes explained in line 34/35 in a schematic figure and include this in the introduction.

In the review articles there are very good schematic figures referring to the deletion, therefore we do not believe it pertinent to include a similar figure in this work (which is not a review). However, we have added a bibliographic reference to a very recent review in which this type of representation appears: Koezel et al, Williams Syndrome Nature Reviews Disease Primers volume 7, Article number: 42 (2021)

3.) Please add genetic cardiomyopathies and especially desminopathies to the list of cardiac diseases associated with mitochondrial dysfunction. For example, it is known, that desminopathy, cause by desmin variants, is also associated with mitochondrial dysfunction. Please add therefore this information including a relevant reference.

We have added the desminopathies to the heart diseases associated with mitochondrial and the representative references (line 52)

4.) Rephrase sentence in Line 98/99. This sounds strange.

We have rephrased the sentence as follow: “Additional mitochondrial damage due to increased ROS/RNS and decreased ATP production via feedback loop results in left ventricular dysfunction”.

5.) Explain all abbreviations like FBS or DMEM in the revised manuscript.

All the abbreviations have been explained:

Dulbecco's Modified Eagle Medium (DMEM

Fetal bovine serum (FBS) 

Phosphate-buffered saline (PBS)

6.) Please present all data as mean with standard deviation SD instead of SEM. Larger error bars are not a problem.

All graphs have been corrected to present data as mean ± SD

7.) Please specify the statistic test, which you used. Non-parametric or parametric tests?

The statistical test used in each analysis is specified in the corresponding figure caption. Additionally, all the results of the statistical analyzes carried out have been included as supplementary material. The respective call to these data is made at the end of the results paragraph of each section.

8.) Figure 3B: Scale bars are missing and not explained.

To better visualize the scale of the images we have enlarged it and the size is indicated in the figure caption

9.) Please specify the exact membranes used for Western blotting.

The exact membranes used for Western blotting has been indicated (line 354)

10.) Please indicate the species used for secondary antibodies?

The species of the secondary antibodies have been indicated in the corresponding section (line 377)

11.) Please include the oligonucleotides used for qRT-PCR.

In line 369, we make the corresponding call to display the sequence of the primers used in the supplementary material (Table S5).

12.) Please include the exact excitation and emission wavelengths used for confocal microscopy.

The exact excitation and emission wavelengths used for confocal microscopy have been indicated in the corresponding section (line 378-380).

Thanks for comments

Reviewer 2 Report

This is a very interesting report on a murine model of the Williams-Beuren Syndrome.   1. Supravalvular should be written without hypen. 2. In general, English language and style should be improved. Revision by a native English speaker is indicated. 3. I do not think that Materials and Methods should be presented as section 4 in IJMS but rather as section 2.  4. No data on cardiac function in this murine model of the Williams-Beuren Syndrome are presented.  5. Is there hypercalcemia in this model of the Williams-Beuren Syndrome?  6. There are several possible congentital anomalies in patients with the Williams-Beuren syndrome. Did the authors study the anatomy in detail?

The structure of many sentences is not adequate.

Author Response

This is a very interesting report on a murine model of the Williams-Beuren Syndrome.  

  1. Supravalvular should be written without hypen.

We have revised all the work to show supravalvular aortic stenosis without hyphen

  1. In general, English language and style should be improved. Revision by a native English speaker is indicated.

English has been edited by a native English speaker (see acknowledgments)

  1. I do not think that Materials and Methods should be presented as section 4 in IJMS but rather as section 2. 

We have followed the template of the journal

  1. No data on cardiac function in this murine model of the Williams-Beuren Syndrome are presented. 

We have included a paragraph in the introduction indicating the cardiovascular alterations described for CD model (lines 92-97)

  1. Is there hypercalcemia in this model of the Williams-Beuren Syndrome?  6. There are several possible congentital anomalies in patients with the Williams-Beuren syndrome. Did the authors study the anatomy in detail?

Regarding characterization of the complete deletion model used in this work, it was described in Segura-Puimedon et al., 2014 (see reference 10). Regarding survival, there were no significant differences between CD and WT animals, as it has been reported in other published partial mouse models for WBS (Li H.H., et al.  Induced chromosome deletions cause hypersociability and other features of Williams-Beuren syndrome in mice, EMBO Mol. Med., 2009, vol. 1(pg. 50-65)). The most common cause of death in CD animals was the development of lymphomas, similar to WT animals of the C57BL/6 background.

Hypercalcemia and congenital abnormalities of the model are not the focus of this work. Transient hypercalcaemia has been documented in approximately 15% of infants and children, although further studies are needed to determine its exact prevalence at all ages. Hypercalcaemia is usually mild and may be accompanied by hypercalciuria in some cases ( Pober BR. Williams-Beuren syndrome. N Engl J Med 2010;362:239–52: Pober BR, Morris CA. Diagnosis and management of medical problems in adults with Williams-Beuren syndrome. Am J Med Genet C Semin Med Genet 2007;145C:280–90). No hypercalcaemia/ hypercalciuria was reported in a metabolic study of WBS mouse models (Palacios-Verdú MG et a., Metabolic abnormalities in Williams–Beuren síndrome. . J Med Genet 2015;52:248–255)

Round 2

Reviewer 2 Report

The authors have provided an ad rem response to the comments of the reviewer.

English language and style have been significantly improved.